# Inferring selection effects in SARS-CoV-2 with Bayesian Viral Allele Selection

**Martin Jankowiak** [1] *, **Fritz H. Obermeyer** [1,2‡], **Jacob E. Lemieux** [1,3]

**1** Broad Institute of Harvard and MIT, Cambridge, Massachusetts, United States of America, **2** Generate Biomedicines, Cambridge, Massachusetts, United States of America, **3** Division of Infectious Diseases, Massachusetts General Hospital, Cambridge, Massachusetts, United States of America

‡Work done at the Broad Institute.
* mjankowi@broadinstitute.org

**Data Availability Statement:** The SARS-CoV-2 data used in our analysis are provided by GISAID. A complete list of accession numbers for the viral genomes used in our study is publicly available: https://github.com/broadinstitute/bvas/raw/main/

## Abstract

The global effort to sequence millions of SARS-CoV-2 genomes has provided an unprecedented view of viral evolution. Characterizing how selection acts on SARS-CoV-2 is critical to developing effective, long-lasting vaccines and other treatments, but the scale and complexity of genomic surveillance data make rigorous analysis challenging. To meet this challenge, we develop Bayesian Viral Allele Selection (BVAS), a principled and scalable probabilistic method for inferring the genetic determinants of differential viral fitness and the relative growth rates of viral lineages, including newly emergent lineages. After demonstrating the accuracy and efficacy of our method through simulation, we apply BVAS to 6.9 million SARS-CoV-2 genomes. We identify numerous mutations that increase fitness, including previously identified mutations in the SARS-CoV-2 Spike and Nucleocapsid proteins, as well as mutations in non-structural proteins whose contribution to fitness is less well characterized. In addition, we extend our baseline model to identify mutations whose fitness exhibits strong dependence on vaccination status as well as pairwise interaction effects, i.e. epistasis. Strikingly, both these analyses point to the pivotal role played by the N501 residue in the Spike protein. Our method, which couples Bayesian variable selection with a diffusion approximation in allele frequency space, lays a foundation for identifying fitness-associated mutations under the assumption that most alleles are neutral.

## Author summary

The SARS-CoV-2 pandemic has been shaped by the repeated emergence of new viral lineages and mutations. Methods to identify emerging variants of epidemiological significance and characterize mutational determinants of enhanced fitness are important for public health. We develop Bayesian Viral Allele Selection (BVAS), a method that leverages the millions of SARS-CoV-2 viral genomes that have been sequenced across the globe to identify mutations linked to increased viral fitness. Ranked lists of top BVAS hits can be used to prioritize lineages and mutations for follow-up study in the lab. By providing a genome-wide view of the evolution of SARS-CoV-2, our principled probabilistic model complements more targeted experimental approaches for elucidating the functional

paper/accession_ids.txt.xz The UShER tree used in our pre-processing pipeline is publicly available: https://hgwdev.gi.ucsc.edu/~angie/9f94a7b/. The vaccination data we use are provided by OWID: https://github.com/owid/covid-19-data/. An open-source implementation of our analysis code is available at: https://github.com/broadinstitute/bvas. The initial portion of our data pre-processing pipeline relies on open source code described by Obermeyer et al. (2022): https://github.com/broadinstitute/pyro-cov Allele-level and lineage-level inference results from our main BVAS analysis are publicly available: https://github.com/broadinstitute/bvas/raw/main/paper/allele_summary.csv https://github.com/broadinstitute/bvas/raw/main/paper/growth_rates_summary.csv.

**Funding:** This work was supported in part by grants from MassCPR Viral Variants Program and CDC BAA 75D30120C09605 (to J.E.L.). - https://www.cdc.gov/amd/whats-new/cdc-announces-awards-SARS-CoV-2-sequencing-SPHERES-initiative.html - https://masscpr.hms.harvard.edu/ The funders had no role in study design, data collection and analysis, decision to publish, or preparation of the manuscript.

**Competing interests:** The authors have declared that no competing interests exist.

consequences of different viral mutations. We argue that running BVAS periodically as part of a real-time genomic surveillance program could provide valuable information for public health authorities about new lineages as they emerge.

# 1 Introduction

The SARS-CoV-2 pandemic has seen the repeated emergence of new viral lineages with higher fitness, where fitness includes any attribute that affects the lineage's growth, including its basic reproduction number and generation time. The virus has evolved into numerous sublineages that are characterized by distinct phenotypes including enhanced pathogenicity, increased escape from convalescent and vaccine-acquired immunity, differential host tropism, and altered biochemical interaction with cell surface machinery. For example, the Spike mutation S:D614G, found in nearly all Variants of Concern, is associated with higher SARS-CoV-2 loads [1, 2]. Other mutations such as S:N439R, S:N501Y, and S:E484K, have been linked, respectively, to increased transmissibility [3], enhanced binding to ACE2 [4], and antibody escape [5, 6]. For the vast majority of observed mutations, however, links to SARS-CoV-2 fitness are unknown and functional consequences remain uncharacterized.

Fortunately, the SARS-CoV-2 pandemic has prompted a global genomic surveillance program of unprecedented scope and scale, with more than 10 million virus genomes sequenced to date. This growing quantity of genomic surveillance data provides a unique opportunity to interrogate the dynamics of viral infection and quantify selective forces acting on lineages and mutations. Current methods to analyze such data typically rely on phylogenetic analysis or parametric growth models. Phylogenetic methods usually rely on computationally expensive Markov Chain Monte Carlo (MCMC) for inference, with the result that handling more than $\sim$5000 samples becomes infeasible [7, 8]. By contrast, parametric growth models are scalable to large datasets but typically do not systematically account for competition between multiple lineages, have little to say about newly emergent lineages, and cannot pinpoint the genetic determinants of differential fitness [9, 10].

Two recently developed methods address some of these shortcomings. The first method, $PyR_0$, is a hierarchical Bayesian parametric growth model that jointly estimates growth rates for multiple lineages across multiple geographic regions [11]. Since $PyR_0$ regresses growth rates against genotype, it can also make inferences about the genetic determinants of differential fitness. Moreover, since $PyR_0$ relies on variational inference it can be applied to large datasets. However, variational inference also results in poor uncertainty estimates and the parametric likelihood that underlies $PyR_0$ makes ad hoc assumptions about the noise characteristics of surveillance data. The second method, which we refer to as MAP, likewise regresses growth rates against genotype but instead utilizes an elegant diffusion-based likelihood that is better suited to the stochastic dynamics of viral transmission [12]. However, unlike $PyR_0$ MAP does not assume that most alleles are approximately neutral, i.e. that most alleles have little or no effect on viral fitness. As a result MAP risks inferring non-negligible selection effects for implausibly many alleles.

In the following we set out to formulate a method—Bayesian Viral Allele Selection—that combines and improves upon the respective strengths of both $PyR_0$ and MAP and achieves the following desiderata. First, we retain both methods' scalability to large datasets and their ability to account for competition between multiple co-circulating lineages. Second, we retain both methods' ability to infer the genetic determinants of differential fitness, which is important both for understanding the biology of transmission and pathogenesis and for predicting the

fitness of emergent lineages. Third, we incorporate the sparsity assumption of PyR$_0$—namely that most alleles are approximately neutral—while adopting the principled diffusion-based likelihood that underlies MAP. Finally we discard variational inference in favor of efficient MCMC so as to obtain more plausible uncertainty estimates, which provide crucial nuance for public health agencies.

To establish the operational characteristics of our method we perform a large suite of simulations, including detailed comparisons to PyR$_0$, MAP, and another diffusion-based method we introduce (Laplace). We find that Bayesian Viral Allele Selection (BVAS) performs well across the board, with notable advantages of BVAS being its robustness to hyperparameter choices, its satisfactory uncertainty estimates and the fact that it offers interpretable Posterior Inclusion Probabilities that can be used to prioritize alleles for follow-up study.

We apply BVAS to 6.9 million SARS-CoV-2 genomes obtained through April 18[th], 2022, noting that, to the best of our knowledge, this is the largest such analysis to date. Our genome wide analysis identifies known functional hot spots in the SARS-CoV-2 genome like the receptor-binding domain (RBD) in the S gene as well as additional hits in regions of the genome whose function is less well understood like the ORF1ab polyprotein. We argue, based on a retrospective backtesting analysis, that running BVAS periodically as part of a real-time genomic surveillance program could provide valuable estimates of the growth rates of new lineages as they emerge. In addition, we conduct an analysis that allows for vaccination-dependent selection effects and find tantalizing evidence that S:N501Y exhibits vaccination-dependent differential fitness. Finally, we conduct an analysis that aims to identify pairs of mutations whose fitness effect is not additive (i.e. epistasis), which likewise points to the important role played by the RBD residue N501.

## 2 Models and methods

### 2.1 Viral infection as diffusion

The starting point for both MAP and BVAS is a branching process that encodes the dynamics of infected individuals at time $t$ stochastically generating secondary infections at time $t + 1$. (A more detailed exposition of this and the following sections can be found in S1 File.) SARS-CoV and SARS-CoV-2 are known to exhibit super-spreading [13, 14] whereby a minority of infected individuals causes the majority of secondary infections. To account for this behavior the number of secondary infections is assumed to be governed by a Negative Binomial distribution, which has a large variance for small values of the dispersion parameter $k$. In particular we assume that if a given individual is infected with a variant $v$ with reproduction number $R_v$, the number of secondary infections due to that individual has mean $R_v$ and variance $R_v + R_v^2/k$. If we let $n_v(t)$ denote the total number of individuals at time $t$ infected with variant $v$, our assumptions result in the following discrete time process:

$$n_v(t+1) \sim \text{NegBin}(\text{mean}=n_v(t)R_v, \text{dispersion}=n_v(t)k) \tag{1}$$

To connect these dynamics to genotype, we assume that variants are characterized by $A$ alleles and that each variant $v$ is encoded as a binary vector $\mathbf{g}_v \in \{0, 1\}^A$. We then express $R_v$ as $R_v = R_0(1 + \Delta R_v)$, where $R_0$ corresponds to the wild-type variant, and assume that $\Delta R_v$ is governed by a linear additive model

$$\Delta R_v = \sum_{a=1}^{A} g_{v,a}\beta_a \tag{2}$$

where $\boldsymbol{\beta} \in \mathbb{R}^A$ are allele-level selection coefficients. Note that we consider quadratic effects in

Sec. 4.7. If we transform from case counts $n_v(t)$ to allele frequencies $x_a(t)$, Lee et al. [12] show that the dynamics in Eq (1) are equivalent to the following diffusion process in allele frequency space

$$\mathbf{x}(t+1) \sim \mathcal{N}(\mathbf{x}(t) + \mathbf{d}(t), v^{-1}\mathbf{\Lambda}(t)) \tag{3}$$

where $\mathbf{d}(t) \in \mathbb{R}^A$ is the $A$-dimensional drift, given by

$$d_a(t) = x_a(t)(1 - x_a(t))\beta_a + \sum_{b \neq a}(x_{ab}(t) - x_a(t)x_b(t))\beta_b \tag{4}$$

The $A \times A$ diffusion matrix $\mathbf{\Lambda}(t)$ is given

$$\Lambda_{ab}(t) = x_{ab}(t) - x_a(t)x_b(t) \tag{5}$$

where $x_{ab}(t)$ is the fraction of infected individuals at time $t$ who carry alleles $a$ and $b$. Finally $v$ is the effective population size given by

$$v \equiv \left(\frac{1}{R_0} + \frac{1}{k}\right)^{-1} n = \frac{kR_0}{k + R_0} n \tag{6}$$

where $n$ is the total number of infected individuals. Importantly, the equivalence of Eqs (1) and (3) holds in the diffusion limit of large $n$.

It is important to note that the use of diffusion processes similar to that in Eq (3) has a long history in population genetics, including seminal work by Kimura [15] as well recent applications that employ diffusion-based likelihoods in the context of statistical inference [16–19].

## 2.2 MAP

The simplest model that utilizes the diffusion-based likelihood in Eq (3) is formulated as follows (we refer the reader to [12] for additional discussion). First we place a Multivariate-Normal prior on the selection coefficients $\boldsymbol{\beta}$

$$p(\boldsymbol{\beta}|\tau) = \mathcal{N}(\boldsymbol{\beta}|\mathbf{0}, \tau^{-1}\mathbb{1}_A) \tag{7}$$

where $\tau > 0$ is the prior precision and $\mathbb{1}_A$ is the $A \times A$ identity matrix. For observed incremental allele frequency changes

$$\mathbf{y}(t) \equiv \mathbf{x}(t+1) - \mathbf{x}(t) \tag{8}$$

the likelihood is given by

$$p(\mathbf{y}_{1:T-1}|\boldsymbol{\beta}, v) = \prod_{t=1}^{T-1} \mathcal{N}(\mathbf{d}(t|\boldsymbol{\beta}), v^{-1}\mathbf{\Lambda}(t)) \tag{9}$$

where we have assumed that $v$ is constant across time. Since $\boldsymbol{\beta}$ appears linearly in the drift $\mathbf{d}(t|\boldsymbol{\beta})$ and the prior is Multivariate-Normal, the corresponding maximum a posteriori (MAP) estimate is available in closed form:

$$\boldsymbol{\beta}^{\text{MAP}} = \left(\sum_{t=1}^{T-1}\mathbf{\Lambda}(t) + \frac{\tau}{v}\mathbb{1}_A\right)^{-1}(\mathbf{x}(T) - \mathbf{x}(1)) \tag{10}$$

An attractive property of this estimator is that it can be computed in $\mathcal{O}(A^3)$ time and is thus quite fast on modern hardware, at least for $A$ up to $A \sim 10^4$–$10^5$. An unattractive property of this estimator is that it can perform poorly in the high-dimensional regime, $A \gg 1$, since we expect most alleles to be neutral, but $\beta_a^{\text{MAP}}$ will generally be non-zero for all $a$.

## 2.3 Bayesian Viral Allele Selection

We now introduce our method: Bayesian Viral Allele Selection (BVAS). We expect most alleles to be nearly neutral ($\beta_a \approx 0$) and we would like to explicitly include this assumption in our model. To do so we utilize the modeling motif of Bayesian Variable Selection [20]:

$$
\begin{aligned}
\text{[inclusion variables]} \qquad & \gamma_a \sim \text{Bernoulli}(h) \\
\text{[selection coefficients]} \qquad & \boldsymbol{\beta}_\gamma \sim \mathcal{N}(0, \tau^{-1}\mathbb{1}_{|\gamma|}) \\
\text{[allele frequency changes]} \qquad & \mathbf{y}(t) \sim \mathcal{N}(\mathbf{d}(t|\boldsymbol{\beta}_\gamma), v^{-1}\boldsymbol{\Lambda}(t))
\end{aligned}
\qquad (11)
$$

where $a = 1, \ldots, A$ and $t = 1, \ldots, T-1$. Here each Bernoulli latent variable $\gamma_a \in \{0, 1\}$ controls whether the $a^{\text{th}}$ coefficient $\beta_a$ is included ($\gamma_a = 1$) or excluded ($\gamma_a = 0$) from the model; in other words it controls whether the $a^{\text{th}}$ allele is neutral or not. The hyperparameter $h \in (0, 1)$ controls the overall level of sparsity; in particular $S \equiv hA$ is the expected number of non-neutral alleles a priori. The $|\gamma|$ coefficients $\boldsymbol{\beta}_\gamma \in \mathbb{R}^{|\gamma|}$ are governed by a Normal prior with precision $\tau$ where $\tau > 0$ is a fixed hyperparameter. Here $|\gamma| \in \{0, 1, \ldots, A\}$ denotes the total number of non-neutral alleles in a given model. Note that in the following we drop the $\gamma$ subscript on $\boldsymbol{\beta}_\gamma$ to simplify the notation.

In addition to inducing sparsity, an attractive feature of the model in Eq (11) is that—because it is formulated as a model selection problem—it explicitly reasons about whether each allele is neutral or not. In particular this model allows us to compute the *Posterior Inclusion Probability* or PIP, an interpretable score that satisfies $0 \le \text{PIP} \le 1$. The PIP is defined as $\text{PIP}(a) \equiv p(\gamma_a = 1|\mathbf{y}_{1:T-1})$, i.e. PIP($a$) is the posterior probability that allele $a$ is included in the model. This quantity should be contrasted to $h$ in Eq (11), which is the *a priori* inclusion probability. Alleles that have large PIPs are good candidates for being causally linked to viral fitness.

In Eq (11) we assume that $h$ is known. An alternative is to place a prior on $h$, $h \sim \text{Beta}(\alpha_h, \beta_h)$, and infer $h$ from data. See Sec. S9 in S1 File for details.

## 2.4 MCMC inference

BVAS admits efficient MCMC inference via a recently introduced algorithm dubbed Tempered Gibbs Sampling [21]. This is quite remarkable: the underlying inference problem is very challenging, since i) it is a transdimensional inference problem defined on a mixed discrete/continuous latent space; and ii) the size of the model space, namely $2^A$, is astronomically large. The feasibility of MCMC inference in this setting is enabled by the specific Gaussian form of the diffusion-based likelihood in Eq (9) and would be impractical for most other (non-conjugate) likelihoods. Thus BVAS is made possible by a pleasant synergy between the form of the prior and the likelihood.

As we explain in more detail in Sec. S4 in S1 File the resulting inference algorithm has $\mathcal{O}(|\gamma|^2 A)$ computational cost per MCMC iteration and is thus quite fast on modern hardware. Here $|\gamma|$ is the total number of non-neutral alleles, which by assumption satisfies $|\gamma| \ll A$. Notably the computational complexity does not include terms that are quadratic or cubic in $A$, since the (strict) sparsity of Bayesian variable selection implies that the required linear algebra never involves $A \times A$ matrices. Importantly, the viability of MCMC inference means that we expect to achieve satisfactory uncertainty estimates, in particular ones that explicitly weigh differing hypotheses about which alleles are neutral and which are not. Indeed the BVAS posterior mean of $\boldsymbol{\beta}$ can be viewed as an evidence-weighted linear combination of $2^A$ MAP estimates.

## 2.5 Multiple spatial regions

In the above we have assumed a single spatial region. To apply either BVAS or MAP to multiple spatial regions we simply add a subscript where necessary and form a product of diffusion-based likelihoods for $N_R$ regions indexed by $r$:

$$\prod_{r=1}^{N_R}\prod_{t=1}^{T-1}\mathcal{N}(\mathbf{d}_r(t|\boldsymbol{\beta}), v_r^{-1}\boldsymbol{\Lambda}_r(t)) \tag{12}$$

As discussed in Sec. S4 in S1 File, including multiple regions has negligible impact on the computational cost, since all summations over the region index $r$ are performed once in pre-processing.

## 2.6 Estimating the effective population size

The likelihood in Eq (9) depends on the effective population size $v$, a quantity that we do not know a priori and need to estimate from data. For a given region $r$ Eq (9) implies

$$\mathbb{E}[\mathbf{y}_r(t)^{\mathrm{T}}\mathbf{y}_r(t)] = \mathbf{d}_r(t)^{\mathrm{T}}\mathbf{d}_r(t) + v_r^{-1}\mathrm{Tr}\boldsymbol{\Lambda}_r(t) \tag{13}$$

so that if we assume that the drift term is subdominant we obtain the approximation

$$\hat{v}_r \approx \frac{\mathrm{Tr}\boldsymbol{\Lambda}_r(t)}{\mathbb{E}[\mathbf{y}_r(t)^{\mathrm{T}}\mathbf{y}_r(t)]} \tag{14}$$

We note that, since $\mathbf{d}_r(t)^{\mathrm{T}}\mathbf{d}_r(t) \geq 0$, we would expect $\hat{v}_r$ to be an underestimate of $v_r$, especially if the effective population size is large. This results in the following simple estimator

$$\hat{v}_r = \frac{1}{T-1}\sum_{t=1}^{T-1}\frac{\mathrm{Tr}\boldsymbol{\Lambda}_r(t)}{\mathbf{y}_r(t)^{\mathrm{T}}\mathbf{y}_r(t)} \tag{15}$$

where we have averaged Eq (14) over $T-1$ time steps.

To accommodate multiple regions we compute $\hat{v}_r$ within each region using Eq (15) and then compute a single global effective population size $\hat{v}$ by computing the median of $\{\hat{v}_r\}$. With this choice all regions contribute equally to the likelihood. See Sec. S6 in S1 File for additional details and discussion.

## 2.7 Sampling rate

As we show in Sec. S8 in S1 File an attractive property of the diffusion process in Eq (3) is that it behaves sensibly in the presence of sampling, i.e. the fact that not all viral sequences are observed in real world datasets. Indeed if sampling is i.i.d. and the sampling rate is $\rho$ with $0 < \rho \ll 1$ then the effect of sampling is to renormalize the effective population size in Eq (6) as

$$v \rightarrow \left(\frac{1}{R_0} + \frac{1}{k} + \frac{2}{\rho}\right)^{-1} n \tag{16}$$

This means that the covariance structure in Eq (3) remains intact, which is important because it is precisely this 2$^{\mathrm{nd}}$ order information that helps BVAS and MAP disentangle driver mutations from passenger mutations. This is reassuring because for SARS-CoV-2, where even the most ambitious surveillance programs satisfy $\rho \ll k$, the effective population size is dominated by the effects of sampling and $v \approx \frac{\rho}{2}n$.

## 2.8 Vaccination-dependent effects

Suppose we know the vaccination rate $0 \leq \phi_r(t) \leq 1$ for a given region $r$. We would like to incorporate this information into our modeling by allowing for vaccination-dependent selection. To do so we write the drift in region $r$ as

$$d_{r,a}(t) = x_{r,a}(t)(1 - x_{r,a}(t))(\beta_a + \phi_r(t)\alpha_a) + \sum_{b \neq a}(x_{r,ab}(t) - x_{r,a}(t)x_{r,b}(t))(\beta_b + \phi_r(t)\alpha_b)$$

where $\boldsymbol{\alpha} \in \mathbb{R}^A$ is a second group of selection coefficients whose strength is modulated by the time- and region-local vaccination rate. In particular $\boldsymbol{\alpha}$ only has a non-negligible effect on infection dynamics when $\varphi_r(t)$ is itself non-negligible. Disentangling the effects of $\boldsymbol{\beta}$ and $\boldsymbol{\alpha}$ is difficult a priori. Our hope, however, is that a Bayesian variable selection approach with robust MCMC inference should be up to the task provided we have enough data. See Sec. S7 in S1 File for additional discussion.

## 2.9 Alternative model: Laplace

Finally we describe the simplest modification of MAP that can account for the expected sparsity of non-neutral alleles. For additional discussion please refer to Sec. S10 in S1 File, where we describe several alternative models that make use of the diffusion-based likelihood in Eq (3). In this approach we place a Laplace prior on $\boldsymbol{\beta}$

$$p(\boldsymbol{\beta}|\sigma^{\text{Laplace}}) = \frac{1}{2\sigma^{\text{Laplace}}}\exp\left(-\frac{\|\boldsymbol{\beta}\|_1}{\sigma^{\text{Laplace}}}\right) \tag{17}$$

where $\|\boldsymbol{\beta}\|_1$ is the $L^1$ norm of $\boldsymbol{\beta}$ and $\sigma^{\text{Laplace}} > 0$ is a hyperparameter that controls the expected level of sparsity. We then define the maximum a posteriori estimate under this Laplace prior:

$$\boldsymbol{\beta}^{\text{Laplace}} \equiv \arg \max_{\boldsymbol{\beta}} p(\boldsymbol{\beta}|\sigma^{\text{Laplace}})p(\mathbf{y}_{1:T-1}|\boldsymbol{\beta}, v) \tag{18}$$

For the sake of precision we should probably refer to MAP as MAP-Gaussian and the approach described here as MAP-Laplace. However, for brevity we instead refer to these methods as MAP and Laplace, respectively. The Laplace estimator cannot be computed in closed form but can be readily approximated with iterative optimization techniques. We will consider Laplace alongside BVAS, MAP, and PyR$_0$ in our simulations, which we turn to next.

# 3 Simulation results

To assess the performance of our method we conduct an extensive suite of simulation-based experiments, including experiments that rely solely on simulated data as well as a semi-synthetic experiment that relies on perturbed SARS-CoV-2 data.

## 3.1 Simulation details

Our simulator closely follows the structure of the discrete time process in Eq (1). The most salient details are as follows (see Sec. S13 in S1 File for details). We include exactly 10 non-neutral alleles of varying effect size, with typical reproduction numbers for variants $v$ ranging between 0.9 and 1.1. In each simulation we consider a given number of $N_R$ regions and $T = 26$ time steps. The initial number of infected individuals at time $t = 1$ within each region is drawn from a Negative Binomial distribution with mean $10^4$. Case counts for $t = 2, \ldots, T$ are determined by the stochastic dynamics in Eq (1) with $k = 0.1$. This value of $k$ is chosen since it is consistent with estimates of the SARS-CoV-2 dispersion parameter [22–25]. These raw counts

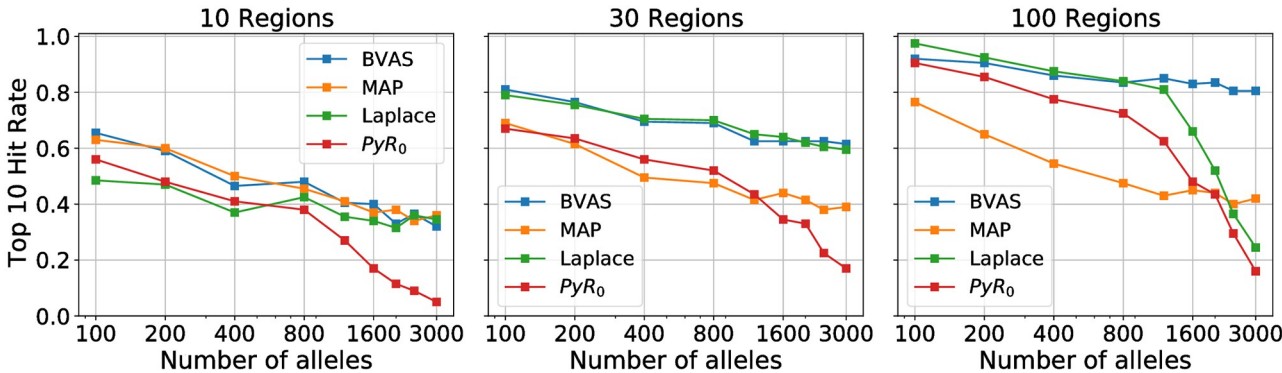

**Fig 1. Hit rate comparison.** We compare the hit rate for four different methods using simulated data, where the hit rate is defined as the fraction of the top 10 hits that are causal. Results are averaged across 20 independent simulations. See Sec. 3.2 for discussion.

are then subjected to Binomial sampling with mean $\rho = 0.01$, i.e. the viral sequences of 99% of cases are not observed. Thus our parameter choices result in simulated data that are highly stochastic and that constitute a regime in which we expect that recovering the true selection coefficients $\boldsymbol{\beta}^*$ is quite challenging. Unless noted otherwise, we generate 20 datasets per condition. We make these choices because they result in simulated data that exhibit some of the characteristics of our SARS-CoV-2 data. In particular, typical estimated effective population sizes $\hat{\nu}$ range from about 25 to about 140 with a mean of about 75.

### 3.2 Method comparison

We compare four methods for inferring allele-level selection using simulated data, in particular three diffusion-based methods (MAP, BVAS, and Laplace) and $PyR_0$. For all methods except for BVAS we rank allele-level hits by the absolute effect size, whereas for BVAS we rank by the Posterior Inclusion Probability (PIP). See Sec. S13.3 in S1 File for the hyperparameter choices made.

In Fig 1 we report results on the hit rate, which we define as the fraction of the top 10 hits that are causal, where causal alleles are those for which the true effect is non-zero. This metric is convenient since it does not depend on any method-specific threshold for calling hits. As expected the hit rate generally increases as the number of regions increases and decreases as the number of alleles increases (since the number of possible spurious hits increases). Strikingly, BVAS exhibits the best hit rates across the board. Laplace and MAP are competitive with BVAS in some regimes, but their performance degrades in other regimes, particularly when the number of alleles is large.

We hypothesize that the main reason for the poor performance of MAP in some regimes is the fact that MAP does not enforce sparsity in the allele-level coefficients $\boldsymbol{\beta}$. This effect is particularly evident from the mean absolute error (MAE) results in Fig 2, where it can be seen that the MAP MAE is large across the board, since MAP assigns non-negligible effect sizes to a large number of alleles. As the number of regions and thus the total amount of data increases, MAP tends to identify ever more non-negligible effects, potentially leading to a large number of spurious hits.

In contrast to MAP, $PyR_0$ and Laplace do impose sparsity on the allele-level coefficients $\boldsymbol{\beta}$. We hypothesize that one of the main reasons for the poor performance of $PyR_0$ and Laplace in some regimes is the fact that they rely on hyperparameters that are difficult to choose. This is

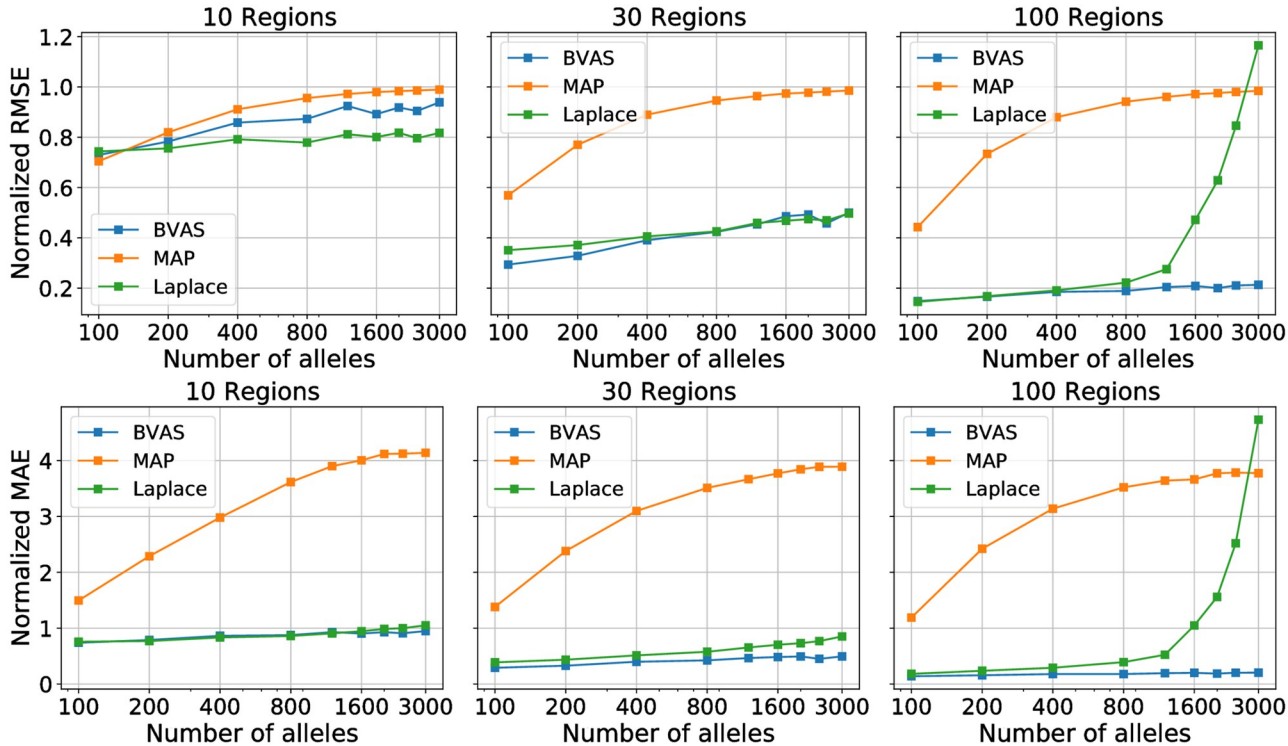

**Fig 2. Coefficient accuracy comparison.** We report the accuracy of inferred selection coefficients $\boldsymbol{\beta}$ for three diffusion-based methods using the simulated data described in Sec. 3.1. We consider two metrics: root mean squared error (RMSE; top) and mean absolute error (MAE; bottom). In both cases the metric is normalized such that the value of the metric for $\boldsymbol{\beta} = \mathbf{0}$ is equal to unity. For example, the RMSE is normalized by $||\boldsymbol{\beta}^*||_2$, where $\boldsymbol{\beta}^*$ are the true effects. We do not include a comparison to $\mathrm{PyR}_0$, since it utilizes a somewhat different likelihood, making direct comparison subtle. See Sec. 3.2 for discussion.

especially the case for $\mathrm{PyR}_0$, which contains 7 model hyperparameters, the most important of which is a direct analog to $\sigma^{\mathrm{Laplace}}$.

To make this broader point concrete we investigate the sensitivity to the Laplace regularization scale $\sigma^{\mathrm{Laplace}}$ in Fig 3. We find that moderate changes in $\sigma^{\mathrm{Laplace}}$ lead to significant degradation in performance. Since there is no principled method to choose $\sigma^{\mathrm{Laplace}}$ a priori, one must instead rely on simulation-based intuition. Since, however, any simulation cannot capture all the effects that characterize real data and since it is unclear a priori what simulation parameters should be used, it remains difficult to choose $\sigma^{\mathrm{Laplace}}$ and so the sensitivity in Fig 3 is troubling. In the next section we show that BVAS exhibits less sensivity to hyperparameter choices.

### 3.3 BVAS sensitivity to hyperparameters and $\hat{v}$

BVAS is specified by two hyperparameters: the prior inclusion probability $h$ and the prior precision $\tau$. For an extended dicussion of $h$ see Sec. S9 in S1 File (note that if a prior is placed on $h$ the hyperparameters instead become $\{\tau, \alpha_h, \beta_h\}$). The quantity $\tau^{-\frac{1}{2}}$ controls the expected scale of effect sizes $\beta$. For example, for $\tau = 100$ the prior standard deviation of $\beta$ is 0.1. This choice implies that $\sim 95\%$ of prior probability mass concentrates on the range $\beta \in [-0.2, 0.2]$. In Fig 4 (top row) we depict the sensitivity of BVAS to changes in $\tau$. We find that the sensitivity to $\tau$ is

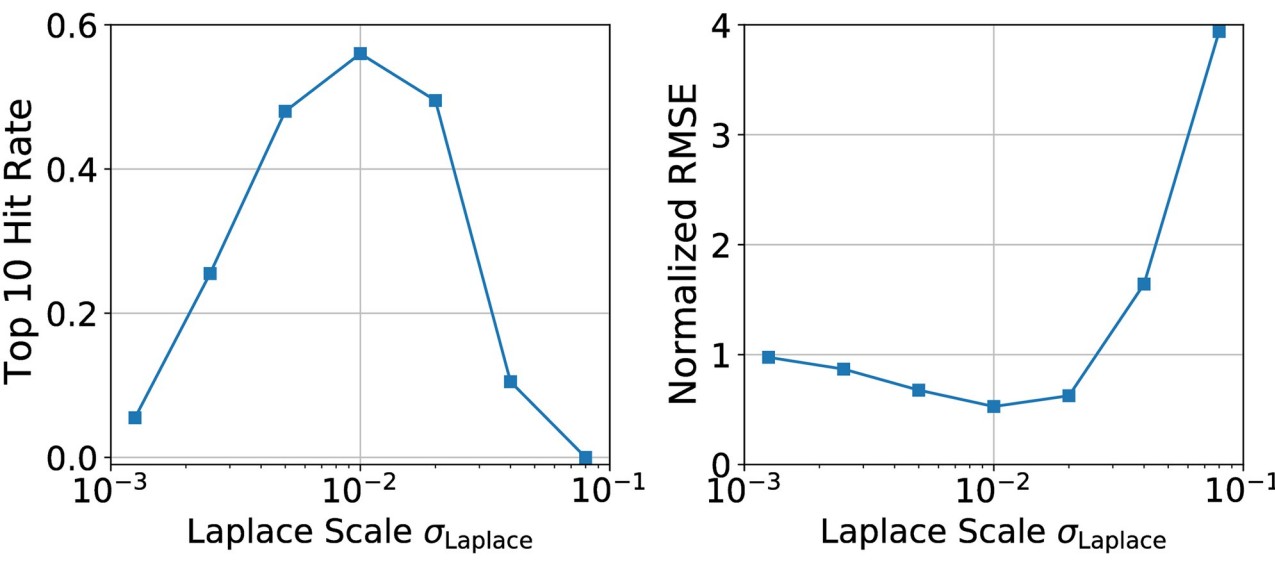

**Fig 3. Sensitivity to the Laplace regularization scale.** We explore the sensitivity of the Laplace method to the prior scale $\sigma^{\text{Laplace}}$. Changing $\sigma^{\text{Laplace}}$ from the optimal value of $\sigma^{\text{Laplace}} \approx 0.01$ results in significantly worse performance. We consider $A = 3000$ alleles and $N_R = 30$ regions and generate 40 simulated datasets.

small over about 4 orders of magnitude. It is only for very large $\tau$ ($\tau = 10^4$) that we see a large drop in performance.

In Fig 4 (bottom row) we depict the sensitivity of BVAS to changes in $S \equiv hA$, which is the expected number of non-neutral alleles a priori. We find only moderate sensitivity as $S$ ranges from $S = 1$ to $S = 64$. In other words it is not necessary for $S$ to be an accurate estimate of the number of non-neutral alleles (10 in our simulations): the posterior is a compromise between the prior and the likelihood and for reasonable choices of $S$ the likelihood can overwhelm the prior if there is sufficient evidence for non-neutral alleles. Importantly the precision remains high for all values of $S$. The effect of choosing small $S$ is to be more conservative; in particular some weak effects at the threshold of discovery may be assigned small PIPs. This robustness to changes in $S$ is reassuring because our a priori knowledge of the number of non-neutral alleles in real data is limited.

Next we explore the sensitivity of BVAS to accurate estimation of the effective population size $\nu$. Note that unlike $S$ or $\tau$, which appear in the prior in Eq (11), $\nu$ appears in the likelihood. The value of $\nu$ evidently plays an important role because it controls the level of noise in the diffusion process. Large values of $\nu$ imply that allele frequency increments $\mathbf{y}(t)$ are largely determined by (deterministic) drift. Conversely, small values of $\nu$ imply that $\mathbf{y}(t)$ exhibits significant (stochastic) variability that dominates the drift. Thus, with all else equal, increasing $\nu$ places more emphasis on fitting the observed apparent drift with the result that BVAS will tend to identify more signal, i.e. more alleles with non-negligible PIPs. Conversely, decreasing $\nu$ places less emphasis on fitting the observed apparent drift with the result that BVAS will tend to identify less signal, i.e. fewer alleles with non-negligible PIPs.

We investigate this effect quantitatively in Fig 5, which confirms our intuition. At least for our simulated data the consequences of underestimating $\nu$ are more severe than the consequences of overestimating $\nu$; for example if we underestimate $\nu$ by a factor of 4 the hit rate drops by a factor of one half. By contrast overestimating $\nu$ by a factor of 4 actually improves the hit rate in this simulation, since the tighter likelihood encourages BVAS to seek out less

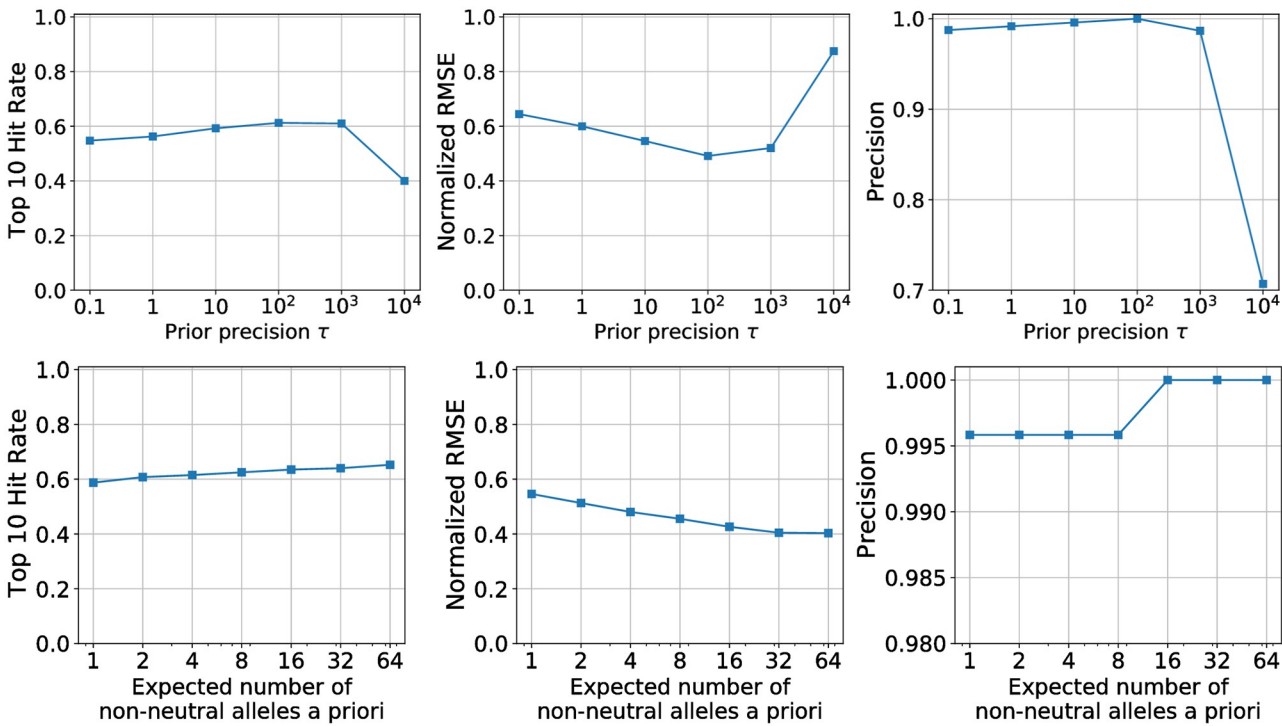

**Fig 4. BVAS hyperparameter sensitivity.** We explore the extent to which BVAS performance is sensitive to its two hyperparameters, namely $\tau$ and $S = hA$, using the simulated data described in Sec. 3.1. We simulate data for $N_R = 30$ regions and $A = 3000$ alleles and generate 40 datasets. See Sec. 3.3 for discussion.

sparse solutions, which results in additional hits for alleles at the margin of discovery. Overall the behavior in Fig 5 is encouraging, since we can estimate the effective population size with moderate accuracy in simulation (see Sec. S13.2 in S1 File). In practice of course we expect worse performance in the context of real data because the noise structure of real data will not precisely follow the noise structure assumed by our diffusion-based likelihood. Nevertheless the fact that the results in Fig 5 exhibit a good degree of robustness for $\nu$ estimates that are off by a factor of $\sim 2$ suggests that running BVAS on real data should be relatively robust to the $\hat{\nu}$ estimation strategy used.

### 3.4 The value of PIPs

In contrast to the other methods we consider BVAS provides a Posterior Inclusion Probability for each allele. In Fig C in S1 File we demonstrate the value of PIPs by exploring the allele-level precision and sensitivity that are obtained if we declare alleles with a PIP above a threshold of 0.1 as hits. We observe very high precision across the board. In other words, if an allele has a high PIP there is good reason to believe it is causally linked to viral fitness, at least if we believe the generative process that underlies our diffusion-based likelihood. It is worth emphasizing that an allele with a moderate effect size can still exhibit a large PIP, thus signifying strong evidence for being causal.

### 3.5 Variability due to sampling rate

As discussed in Sec. 2.7 our diffusion-based likelihood, Eq (9), naturally accommodates random sampling where only a fraction $\rho$ of infected individuals have their viral genomes

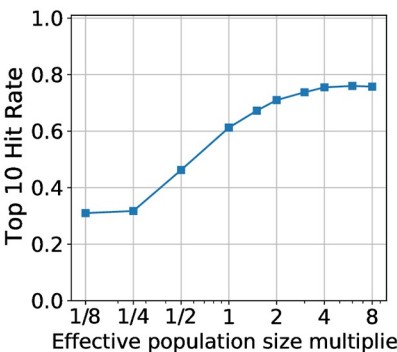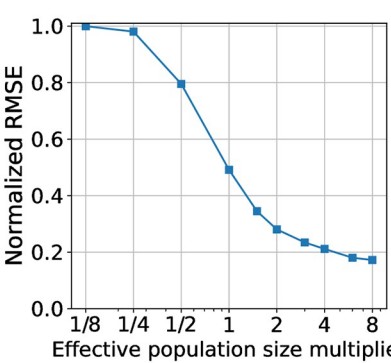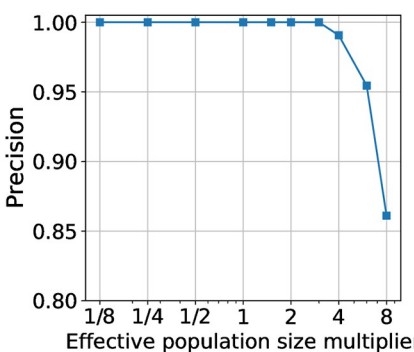

**Fig 5. BVAS effective population size sensitivity.** We explore the extent to which BVAS performance is sensitive to accurate estimation of the effective population size $\nu$ using the simulated data described in Sec. 3.1. To do so we modulate our estimate for $\nu$ by the indicated multiplier, e.g. $\hat{\nu} \rightarrow 2\hat{\nu}$. We simulate data for $N_R = 30$ regions and $A = 3000$ alleles and generate 40 datasets. See Sec. 3.3 for discussion.

sequenced. To explore the effects of sampling we generate data with a sampling rate that ranges between 1% and 64%. We find that the results are remarkably robust (see Fig D in S1 File), even as the effective population size decreases by a factor of $\sim 15$ as $\rho$ decreases from 64% to 1% (see Eq (16)).

## 3.6 Including vaccination-dependent effects

We now incorporate vaccination rates $\phi_r(t)$ into our simulations, assuming that $\phi_r(t)$ starts at zero everywhere and increases linearly over time. We assume 20 non-zero effects, half of which are vaccination-dependent. Otherwise our simulation follows the specifications of Sec. 3.1. See Fig E in S1 File for results. As we would expect, robustly identifying causal mutations is harder in this setting, and the hit rate for vaccination-dependent effects is lower than for all effects. Nevertheless the precision is high in all cases, which gives us confidence that high PIP vaccination-dependent alleles identified in real data may be causally linked to vaccination-dependent differential fitness.

## 3.7 Spike-in experiment

We conduct a semi-synthetic experiment where we add 200 synthetic (and therefore spurious) alleles to 3000 SARS-CoV-2 lineages (we use data from January 20[th] 2022 for a total of $A = 2904 + 200 = 3104$ alleles). Each lineage is assigned a Binomial number of non-wild-type spiked-in alleles with mean 2. Since these assignments are independent and identically distributed, the spiked-in alleles are not correlated with the pre-existing genotype in any way and thus any apparent selection effects due to these alleles are due to chance alone. See Fig 6 for results. We find that PyR$_0$ and BVAS select the fewest number of spiked-in alleles, whereas MAP and Laplace select the most. Note that we expect some small number of spiked-in alleles to be selected due to random chance alone. Importantly, across 30 replications none of the methods identifies a single spiked-in allele in the top 20 scoring hits. This is encouraging, since it suggests that the top scoring hits from all four methods should be enriched with causal alleles.

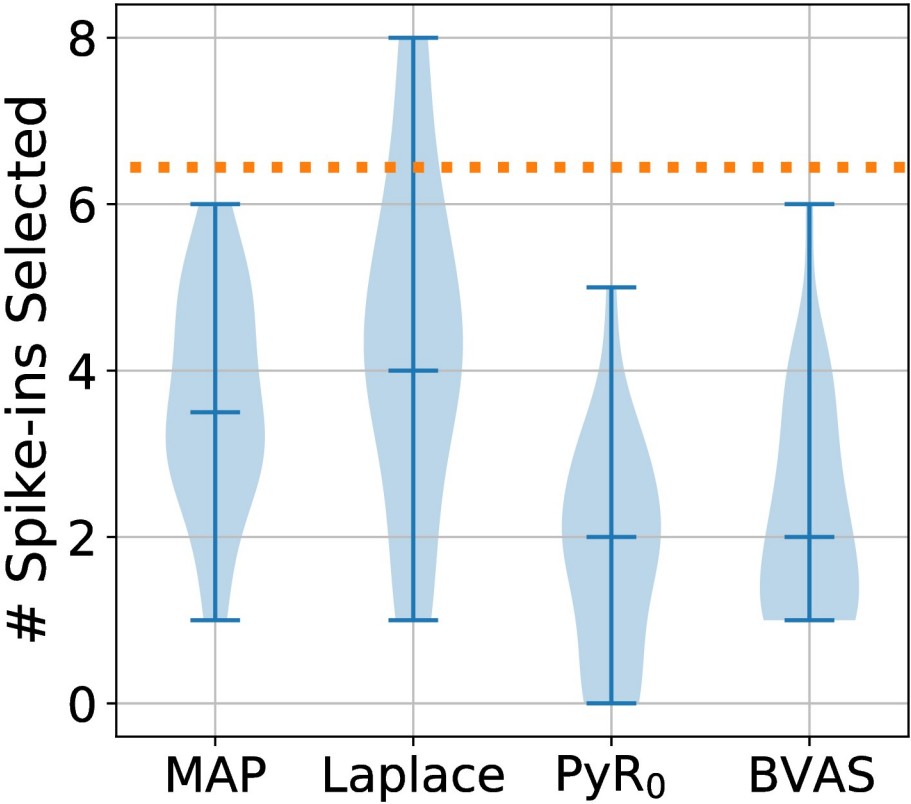

**Fig 6. Semi-synthetic spike-in experiment.** We compare the robustness of four methods for inferring allele-level selection effects to the addition of spiked-in alleles. We depict the total number of spiked-in alleles that are among the top 100 scoring alleles, where the horizontal line within each violin plot denotes the median and for consistency we rank alleles by the absolute value of the selection coefficient $\beta$. The orange dotted line corresponds to the number of spiked-in alleles that would be expected among the top 100 alleles if alleles were ranked at random. We report results from 30 independent simulations.

## 4 SARS-CoV-2 analysis

### 4.1 Data

Our raw data consist of 8.6 million samples downloaded from GISAID [26] on April 18[th], 2022. In initial pre-processing we follow the procedure in Obemeyer et al. [11], which relies on a phylogenetic tree constructed by UShER [27, 28], and results in $L = 3000$ SARS-CoV-2 clusters that are finer than the 1662 PANGO lineages in the data [29]. In our main analysis we consider $A = 2975$ non-synonymous amino acid substitutions, excluding both insertions and deletions due to limitations of UShER, and taking Wuhan A as the reference genotype, i.e. $R_0 \equiv R_A$. After filtering to well-sampled regions there remain 6.9 million samples from $N_R = 128$ regions. Allele frequencies for each region are computed in time bins of 14 days and the effective population size is estimated using the global strategy described in Sec. 4.6. Vaccination data for the analysis in Sec. 2.6 are obtained from OWID [30]. For additional details on data pre-processing see Sec. S14.1 in S1 File. For additional results using data obtained through August 10[th] 2022 see Tables C and D in S1 File.

**Table 1. Top 20 fittest SARS-CoV-2 lineages.**

| Rank | Lineage | Growth Rate | Rank | Lineage | Growth Rate |
|---|---|---|---|---|---|
| 1 | BA.2.12.1 | 7.858 ± 0.727 | 11 | BA.2.4 | 7.399 ± 0.657 |
| 2 | BA.2.11 | 7.835 ± 0.667 | 12 | BA.2.7 | 7.398 ± 0.657 |
| 3 | BA.5 | 7.764 ± 0.688 | 13 | BA.2.6 | 7.398 ± 0.657 |
| 4 | BA.2.12 | 7.599 ± 0.703 | 14 | BA.2.5 | 7.398 ± 0.657 |
| 5 | BA.2.9.1 | 7.400 ± 0.659 | 15 | BA.2.1 | 7.398 ± 0.656 |
| 6 | BA.2.3.2 | 7.400 ± 0.657 | 16 | BA.2.13 | 7.398 ± 0.659 |
| 7 | BA.2.3.1 | 7.399 ± 0.657 | 17 | BA.2.2 | 7.394 ± 0.659 |
| 8 | BA.2.8 | 7.399 ± 0.657 | 18 | BA.2 | 7.370 ± 0.681 |
| 9 | BA.2.3 | 7.399 ± 0.656 | 19 | BA.2.15 | 7.339 ± 0.677 |
| 10 | BA.2.14 | 7.399 ± 0.657 | 20 | BA.2.16 | 7.339 ± 0.676 |

The 20 SARS-CoV-2 lineages with the highest (relative) growth rates $R_v/R_A$ as estimated by BVAS. Here and elsewhere uncertainty estimates are 95% credible intervals.

## 4.2 Fitness of SARS-CoV-2 lineages and mutations

We use BVAS to rank the relative fitness of all SARS-CoV-2 lineages. To do so, we fit our model to allele frequencies of 2975 alleles across 128 regions, with $\tau = 100$ and $S = 50$ (so $h = S/A \approx 0.017$) reflecting our prior assumptions that a relatively modest number of non-neutral alleles with (possibly) moderately large selection effects are driving evolution of SARS-CoV-2 fitness.

In Table 1, we report relative growth rate estimates $R/R_A$ for the top 20 lineages. Fitness estimates are broadly concordant with the observed pandemic, with the fittest lineages all Omicron variants. BVAS accurately captures the hierarchy of replacement by fitter lineages with Omicron (BA.2) > Omicron (BA.1) > Delta > Alpha > wild-type virus (Table 2). Notably some PANGO lineages (e.g. B.1.1) exhibit very diverse genotypes and thus correspondingly diverse growth rates, see Fig G in S1 File.

Analysis of sublineage fitness reveals that Omicron has fractured into many sublineages whose fitness has increased modestly over time (Table 1). BA.2.12.1 appears to be the fittest lineage observed to date, although many BA.2 sublineages are comparably fit. BA.5, which appears to be descended from BA.2, has recently emerged in South Africa and is reported to

**Table 2. Growth rates for selected SARS-CoV-2 lineages.**

| Lineage | WHO Classification | Growth Rate |
|---|---|---|
| BA.2 | Omicron | 7.370 ± 0.681 |
| B.1.1.529 | Omicron | 6.137 ± 1.259 |
| BA.1 | Omicron | 5.752 ± 0.683 |
| B.1.617.2 | Delta | 3.503 ± 0.548 |
| B.1.621 | Mu | 2.909 ± 0.206 |
| P.1 | Gamma | 2.495 ± 0.189 |
| B.1.1.7 | Alpha | 2.262 ± 0.230 |
| B.1.351 | Beta | 2.135 ± 0.171 |
| B.1.1 | | 1.947 ± 1.544 |
| B.1 | | 1.634 ± 0.844 |

We report the relative growth rates $R/R_A$ of selected SARS-CoV-2 lineages as estimated by BVAS. Uncertainty estimates are 95% credible intervals.

**Table 3. Top 25 fitness-associated SARS-CoV-2 mutations.**

| Rank | Mutation | RBD | PIP | Beta | Rank | Mutation | RBD | PIP | Beta |
|---|---|---|---|---|---|---|---|---|---|
| 1 | S:R346K | Yes | 1.000 | 0.371 ± 0.048 | 1 | S:L452R | Yes | 1.000 | 0.435 ± 0.103 |
| 2 | S:L452R | Yes | 1.000 | 0.435 ± 0.103 | 2 | S:E484K | Yes | 1.000 | 0.386 ± 0.097 |
| 3 | S:E484K | Yes | 1.000 | 0.386 ± 0.097 | 3 | S:R346K | Yes | 1.000 | 0.371 ± 0.048 |
| 4 | S:N501Y | Yes | 1.000 | 0.364 ± 0.118 | 4 | S:N501Y | Yes | 1.000 | 0.364 ± 0.118 |
| 5 | M:I82T | | 1.000 | 0.335 ± 0.068 | 5 | S:D614G | | 1.000 | 0.339 ± 0.119 |
| 6 | S:D614G | | 1.000 | 0.339 ± 0.119 | 6 | S:P681R | | 1.000 | 0.338 ± 0.146 |
| 7 | S:A222V | | 1.000 | 0.170 ± 0.058 | 7 | M:I82T | | 1.000 | 0.335 ± 0.068 |
| 8 | S:P681H | | 1.000 | 0.216 ± 0.103 | 8 | S:H655Y | | 0.994 | 0.328 ± 0.154 |
| 9 | S:S477N | Yes | 1.000 | 0.231 ± 0.146 | 9 | S:N679K | | 0.950 | 0.321 ± 0.233 |
| 10 | S:P681R | | 1.000 | 0.338 ± 0.146 | 10 | S:N440K | Yes | 0.999 | 0.296 ± 0.126 |
| 11 | ORF1b:K1383R | | 0.999 | -0.106 ± 0.045 | 11 | S:N764K | | 0.833 | 0.267 ± 0.302 |
| 12 | S:N440K | Yes | 0.999 | 0.296 ± 0.126 | 12 | S:L452Q | Yes | 0.883 | 0.259 ± 0.243 |
| 13 | ORF1b:H1087Y | | 0.998 | 0.165 ± 0.057 | 13 | S:Q954H | | 0.789 | 0.252 ± 0.315 |
| 14 | ORF1a:A2529V | | 0.998 | 0.097 ± 0.039 | 14 | S:N969K | | 0.789 | 0.250 ± 0.315 |
| 15 | S:H655Y | | 0.994 | 0.328 ± 0.154 | 15 | S:S371F | Yes | 0.806 | 0.245 ± 0.289 |
| 16 | ORF1a:A2554V | | 0.954 | -0.115 ± 0.060 | 16 | N:D3L | | 0.914 | 0.238 ± 0.166 |
| 17 | S:N679K | | 0.950 | 0.321 ± 0.233 | 17 | M:A63T | | 0.813 | 0.236 ± 0.263 |
| 18 | ORF1a:T3255I | | 0.947 | 0.196 ± 0.145 | 18 | S:T859N | | 0.945 | 0.236 ± 0.182 |
| 19 | S:T859N | | 0.945 | 0.236 ± 0.182 | 19 | S:S477N | Yes | 1.000 | 0.231 ± 0.146 |
| 20 | N:D3L | | 0.914 | 0.238 ± 0.166 | 20 | S:P681H | | 1.000 | 0.216 ± 0.103 |
| 21 | S:L452Q | Yes | 0.883 | 0.259 ± 0.243 | 21 | ORF1b:I1566V | | 0.698 | 0.214 ± 0.321 |
| 22 | S:N764K | | 0.833 | 0.267 ± 0.302 | 22 | ORF1a:P3395H | | 0.696 | 0.212 ± 0.321 |
| 23 | M:A63T | | 0.813 | 0.236 ± 0.263 | 23 | E:T9I | | 0.693 | 0.208 ± 0.287 |
| 24 | S:S371F | Yes | 0.806 | 0.245 ± 0.289 | 24 | S:S704L | | 0.763 | 0.200 ± 0.257 |
| 25 | S:N969K | | 0.789 | 0.250 ± 0.315 | 25 | S:V213G | | 0.666 | 0.198 ± 0.316 |

The top 25 fitness-associated SARS-CoV-2 mutations as estimated by BVAS and ranked by PIP (left) and $\beta$ (right). The two rankings are largely the same, with 19 mutations appearing in both. Uncertainty estimates are 95% credible intervals.

have enhanced fitness [31] and additional immune escape [32, 33] relative to Omicron BA.1. Since BVAS regresses growth rate against genotype, it is able to infer that BA.5 is among the fittest lineages circulating despite the fact that very few BA.5 sequences are in our dataset. Like BA.2.12.1, BA.5 also possess mutations at Spike position 452 (L452R), underscoring the key role that this site plays in SARS-CoV-2 fitness.

We report the fitness of recombinant lineages in Table A in S1 File. In contrast to highly fit lineages that emerged in the BA.2 clade, several recombinants, including those that represent recombination between Delta and BA.1 and BA.1 and BA.2, have been the source of international concern [34–36]. The fittest recombinants are XN and XT, though their fitness is intermediate to that of BA.2 and BA.1. While the appearance of recombinant lineages is striking, the fitness of existing XA—XT recombinants suggests that these particular lineages are unlikely to play an important role in the future.

We report the fitness of top-scoring mutations in Table 3 and plotted along the length of the genome in Fig 7. The strongest signal of selection is in Spike (see Fig 8), with the greatest concentration of hits located in the receptor-binding domain (RBD). Strong signals of selection are also observed in the N-terminal domain (NTD) and furin cleavage sites. By effect size, S:L452R is the top-scoring hit. This mutation is found in BA.4/BA.5 and was also an important

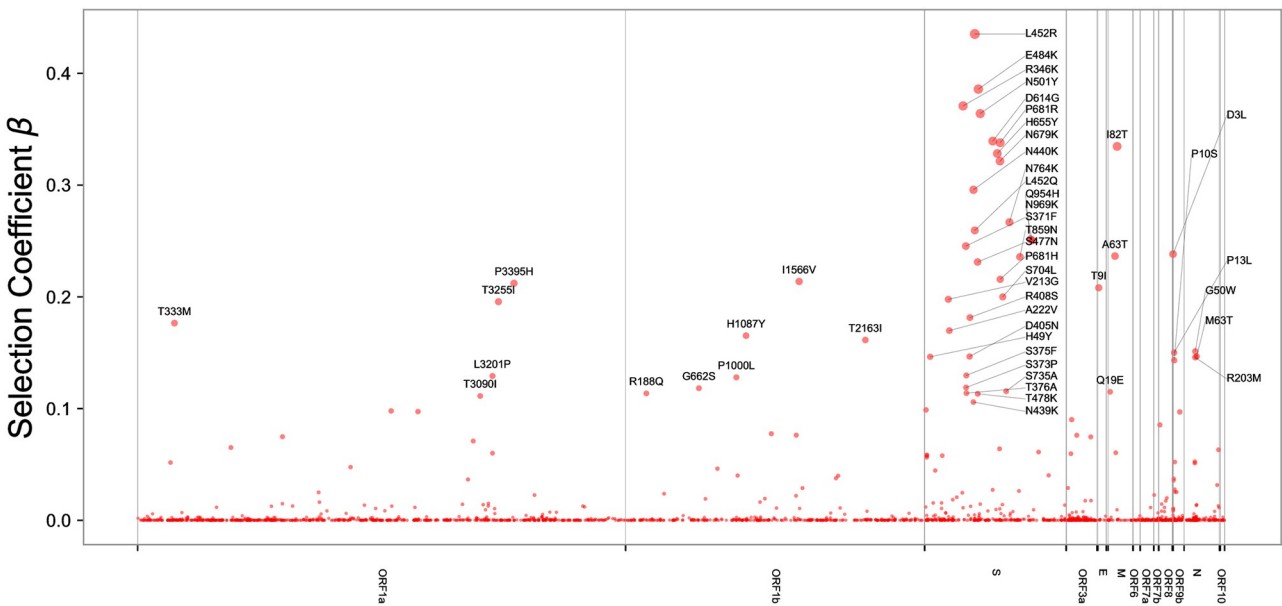

**Fig 7. Genome-wide Manhattan plot.** We depict BVAS hits as a Manhattan plot of the entire SARS-CoV-2 genome, with gene boundaries indicated on the horizontal axis. We do not include the small number of hits with negative selection coefficients (in particular ORF1b:K1383R, ORF1a:A2554V, ORF1b:S959P, and ORF1a:G697R).

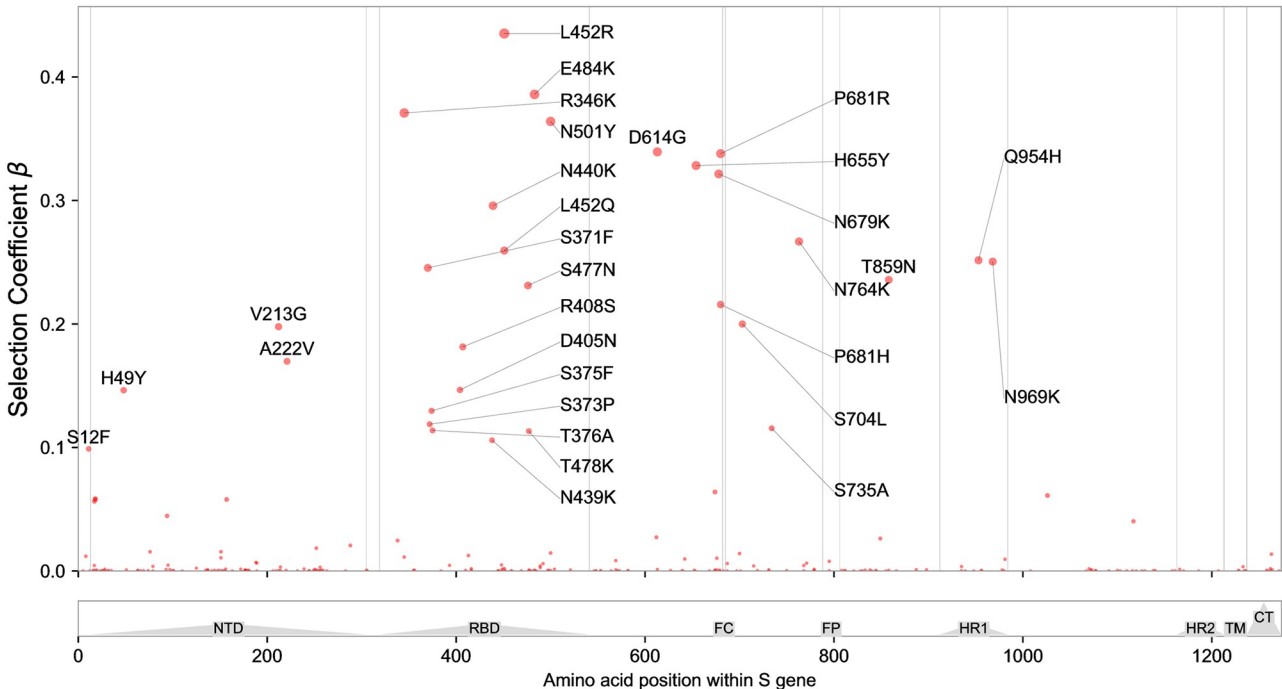

**Fig 8. S-gene Manhattan plot.** View of the 1237 amino acids of the S protein, annotated by structure [47]; many top-scoring mutations are located in the N-terminal domain (NTD), receptor-binding domain (RBD), and furin cleavage (FC) site. Regions containing the fusion peptide (FP), heptad repeat (HR) 1 and 2, transmembrane domain (TM), and C-terminal domain (CTD) are also annotated.

component of the 'California' variants, B.1.427 and B.1.429. Deng et al. [3] have shown that this mutation increases infectivity, while [37] and [38] have shown that it promotes antibody escape. The closely related mutation S:L452Q, one of two key Spike mutations in the fast-growing BA.2.12.1 variant, is also highly ranked, underscoring the importance of this site.

The top-scoring mutations cluster together in various regions of Spike (Fig 9A), particularly the ACE2 binding interface of the RBD (Fig 9B). Three of the top five hits (S:L452R, S:E484K, and S:N501Y) are within the RBD, and a fourth, S:R346K, is just adjacent to it. The mechanisms driving positive selection at these sites likely include increased affinity to ACE2, as was shown for S:N501Y [4], as well as escape from neutralizing antibodies that bind in this region, e.g. for S:E484K, S:N440K and S:R346S [39, 40].

We characterized antibody escape of Spike mutations further by correlating BVAS RBD estimates to predictions made by the antibody-escape calculator in Greaney et al. [41] (Fig H in S1 File). This escape calculator is based on deep mutational scanning data for 33 neutralizing antibodies elicited by SARS-CoV-2 and thus represents an independent source of experimental data. As in [11], there is a strong correlation ($\rho_{Spearman} = 0.89$) between the two sets of predictions, lending support to BVAS results for the RBD. We assessed the temporal progression of selection effects in SARS-CoV-2 lineages by aggregating $\Delta R$ estimates due to S gene, RBD, and non-S-genes contributions (Fig 10). The elevated contribution of S-gene mutations (notably in the RBD) over non-S-gene mutations starting around November 2021 is apparent, in agreement with the results from [11]. Collectively these two results suggest that immune escape has become an increasingly prominent factor in SARS-CoV-2 evolution over time, likely a result of rising rates of convalescent and vaccine-induced immunity to Spike. The correlation between mutations that confer antibody escape and mutations associated with fitness

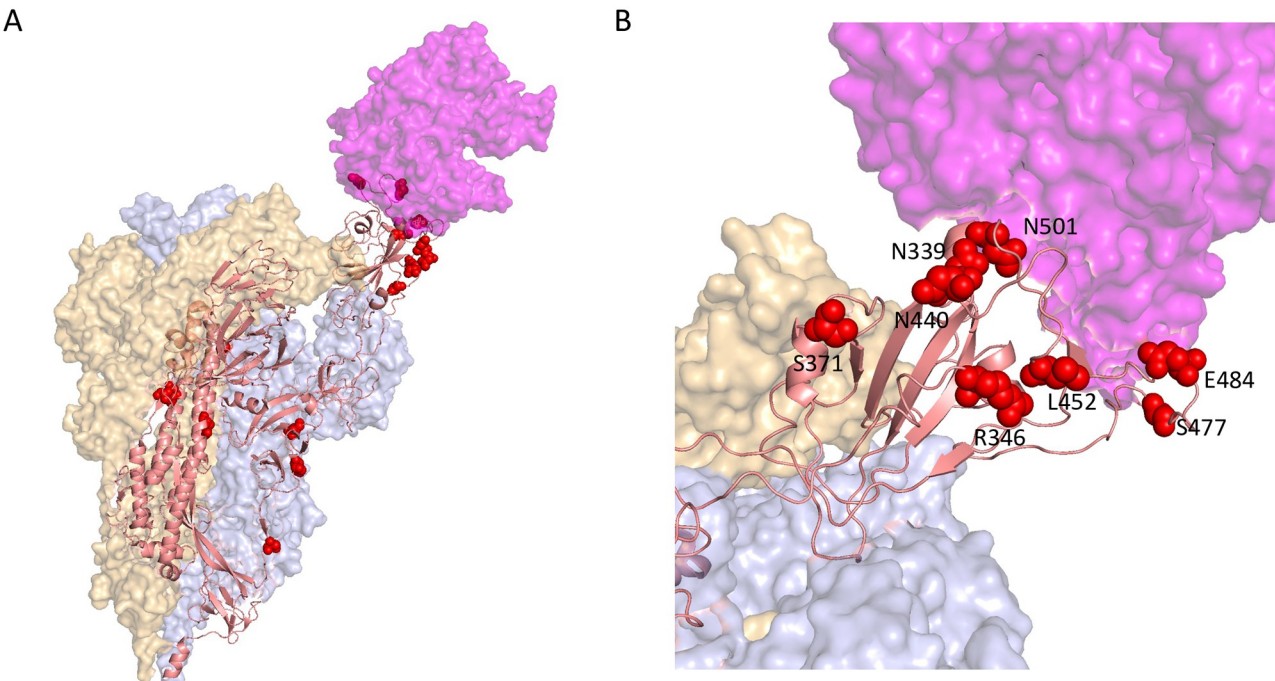

**Fig 9. Top Spike hits in 3D. A**. We depict the locations of the top 20 Spike hits, ranked by PIP, on the Cryo-EM structure of a Spike trimer bound to ACE2 (magenta) at 3.9 Angstrom resolution in the single RBD 'up' conformation from [48] **B**. Enlarged view of the RBD-ACE2 interface, showing the spatial proximity of S:R346, S:N339, S:N440, S:L452, S:S477, S:E484, and S:N501.

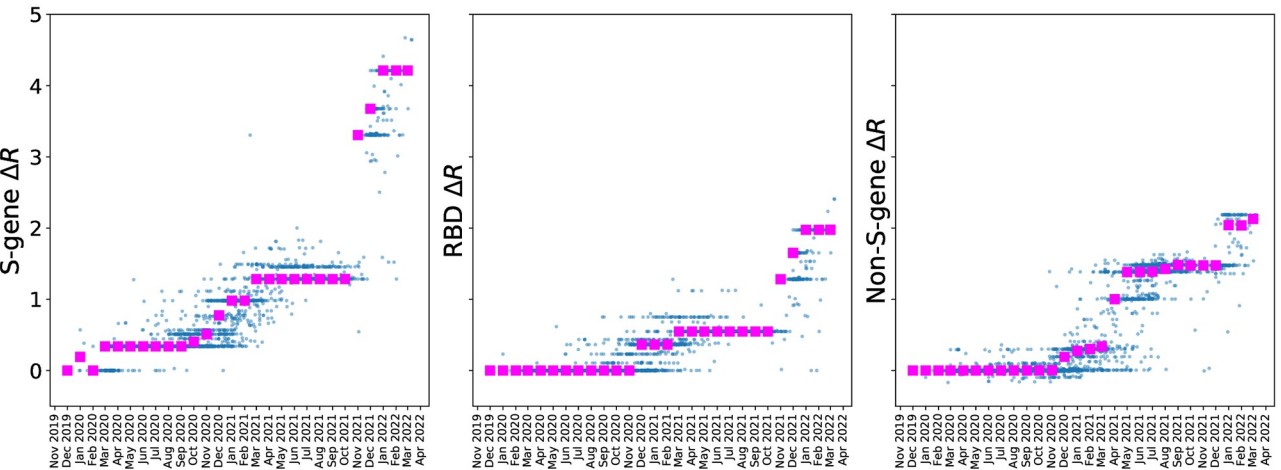

**Fig 10. SARS-CoV-2 'phase diagram'.** We aggregate BVAS $\beta$ estimates into S-gene (including the RBD), RBD (S gene receptor-binding domain), and non-S-gene components for $L = 3000$ SARS-CoV-2 clusters (blue points). The horizontal axis denotes the date at which each cluster first emerged, and magenta squares denote the median $\Delta R$ within each monthly bin. The elevated contribution of S-gene mutations (notably in the RBD) over non-S-gene mutations starting around November 2021 is conspicuous. Compare to Fig 2CDE in [11].

supports the hypothesis that coronaviruses evolve through positive selection of receptor-binding domain mutations that escape neutralizing antibody responses [42].

Other hotspots within Spike include the furin cleavage site, which features three of the top 20 mutations (S:P681R, S:P681H, and S:N679K). These substitutions add positive charge at the cleavage site, likely facilitating S1/S2 cleavage and promoting infectivity by enhancing cell-cell fusion, which has been demonstrated by several groups [43–45].

BVAS also identifies numerous residues under selection outside of Spike. These include P13L in the N-terminal region and P199L and R203M in the linker region between the N- and C-terminal domains (Fig I in S1 File). These latter two mutations have been demonstrated by [46] to enhance viral packaging. Within ORF1a, several amino acid changes in NSP4 score highly, including T3090I, L3201P, and T3255I (Fig J in S1 File). The importance of these effects can be quantified on a per-ORF basis using PIPs, which provide a convenient numerical measure of the total evidence for selection due to each allele. To assess the overall contribution of various regions of the SARS-CoV-2 genome to differential viral fitness we sum PIPs across different regions (see Fig K in S1 File), quantifying the relative importance of S, N, and several non-structural proteins in ORF1ab.

### 4.3 Sensitivity analysis

We perform an extensive sensitivity analysis to better understand the robustness of our results to hyperparameter and data pre-processing choices. In Figs M-Q in S1 File we explore the sensitivity of BVAS growth rate estimates for the fittest lineages, noting that we might expect these estimates to be sensitive to the strength of regularization. We find minimal sensitivity to $S$ (the number of non-neutral alleles expected a priori), the total number of regions included in the analysis $N_R$, as well as $N_{\min}^{14}$, which controls which time bins enter the analysis, see Figs M-O in S1 File. The sensitivity to the prior precision $\tau$ is somewhat larger (see Fig P in S1 File), especially as we increase $\tau$ to $\tau = 400$, although we would argue that this is an unreasonable prior choice, as it makes even relatively moderate selection effects like $\beta \sim 0.15$ a priori unlikely.

Not surprisingly, the sensitivity to the estimated effective population size, $\hat{\nu}$, is fairly large (see Fig Q in S1 File), roughly comparable to the scale of the underlying statistical uncertainty.

We adopt a more global view in Figs R-AA in S1 File, reporting sensitivity analyses for all PIPs and $\beta$ estimates as well as for growth rates for all 1662 PANGO lineages. Globally, growth rate estimates are remarkably stable with Pearson correlation coefficients of 0.999 or larger. Selection coefficient estimates $\beta$ are also quite stable, with Pearson correlation coefficients of 0.9 or greater, with the largest sensitivity being again to $\tau$ and $\hat{\nu}$. Results for allele-level PIPs are broadly comparable, although they tend to exhibit more variability and outliers, especially for alleles with smaller PIPs. Importantly concordance between independent MCMC chains is exceptionally high ($R \geq 0.9997$), see Fig R in S1 File, suggesting that MCMC error is minimal.

### 4.4 Backtesting

We perform a backtesting analysis in which we run BVAS on subsets of the data defined by varying end dates. Doing so allows us to assess the possible benefits of running BVAS periodically as part of a real-time genomic surveillance program. We caution that these results need to be interpreted with care, since in practice it can take several weeks before individual genomic samples are deposited in GISAID. Additionally, the calling of new lineages is also associated with a time lag. See Fig BB in S1 File for results. We find that by May 13$^{\text{th}}$ 2021 BVAS predicts that various Delta sublineages are fitter than B.1.1.7 (Alpha), which was the most prevalent lineage in England and elsewhere at the time. Similarly, we find that by December 8$^{\text{th}}$ 2021 BVAS predicts that various Omicron sublineages are fitter than AY.4.2.1 (Delta). Notably, since BVAS regresses $R_\nu$ against genotype, we also obtain plausible estimates for newly emergent lineages that have only been sequenced a small number of times. Finally by January 12$^{\text{th}}$ 2022 BVAS predicts that BA.2 was fitter than BA.1, a prediction that has been borne out by the subsequent takeover of BA.2.

During the time periods considered in our backtesting analysis the number of samples of these newly emergent lineages was increasing rapidly, with the result that BVAS growth rate estimates increase markedly as more data become available and it becomes increasingly clear that these lineages were the fittest lineages yet observed. Importantly, estimates stabilize after sufficient data have been collected, see Fig CC in S1 File.

### 4.5 Comparison to MAP and PyR$_0$

We perform detailed comparisons of BVAS to both MAP [12] and PyR$_0$ [11]. Here we provide a brief summary; see Sec. S14.5–6 in S1 File and Fig DD-NN in S1 File for a detailed discussion. Despite differences in methodology, results from BVAS, MAP and PyR$_0$ are in broad *qualitative* agreement, suggesting that all three methods are capable of identifying (at least some) leading driver mutations in SARS-CoV-2. Because it is only very recently that genomic surveillance data have become available at this scale and the corresponding analysis methods are still in their infancy, we believe this finding is encouraging for this emerging new field.

While it is difficult to definitively establish the superiority of one method over another without a larger corpus of experimental data to serve as ground truth, we believe that the advantages of BVAS become apparent upon closer comparison. First, inferred selection effects are much sparser in the case of BVAS, which aids interpretability and is arguably more plausible a priori. Selection coefficients inferred by MAP are very dense unless the regularization parameter $\gamma_{\text{reg}} = \tau / \nu$ is made sufficiently large. However, in this limit growth rate estimates appear to be over-regularized towards Wuhan A. Second, uncertainty estimates for MAP and (especially) PyR$_0$ are much narrower than for BVAS, especially for allele-level quantities like selection coefficients. Finally, BVAS exhibits much less sensitivity to hyperparameter choices than

**Table 4. Top vaccination-dependent hits.**

| Mutation | Alpha PIP | Alpha | Beta PIP | Beta |
|---|---|---|---|---|
| S:N501Y | 0.606 | -0.149 ± 0.266 | 1.000 | 0.408 ± 0.121 |
| ORF1a:A2554V | 0.404 | -0.080 ± 0.200 | 0.549 | -0.066 ± 0.122 |

We report PIPs and estimated coefficients $\alpha$ for vaccination-dependent effects for an analysis in which the vaccination status is 'fully-vaccinated'. We report effects that are ranked in the top 75 by PIP. Estimated $\beta$ coefficients and PIPs for the corresponding vaccination-independent effects are reported in the two rightmost columns.

PyR$_0$, which—together with rigorous sparsity requirements—is one of the key factors contributing to the strong performance of BVAS in simulations.

## 4.6 Vaccination-dependent effects

We incorporate independent data on vaccination rates from 127 regions compiled by OWID (see Sec. S7 and Sec. S14.1.6 in S1 File for additional details). This has little impact on estimates for most vaccination-independent effects (Fig PP in S1 File), i.e. BVAS finds that selection effects in the data are largely explainable by vaccination-independent effects. Indeed we find only two alleles with large PIPs in the vaccination-dependent model (Table 4).

The strongest evidence for allele-level dependence on vaccination rates is in S:N501Y, which is also the only allele for which BVAS finds it important to include both a vaccination-dependent effect $\alpha$ *and* a vaccination-independent effect $\beta$. In our analysis the selection coefficient estimates for S:N501Y are $\alpha \approx -0.15$ and $\beta \approx 0.41$. This means that the model predicts a selection effect due to the S:N501Y allele of $\beta \approx 0.41$ in a completely unvaccinated population and a selection effect of $\alpha + \beta \approx 0.26$ in a completely vaccinated population. This can be interpreted as saying that vaccination appears to confer differential protection against the S:N501Y allele, i.e. on top of the protection that is conferred against typical SARS-CoV-2 variants that do not carry S:N501Y. Notably, S:N501Y also exhibits a large PIP in an analysis conducted with a different definition of vaccination rate (Table E in S1 File).

These results are also supported by a direct analysis of raw allele frequencies of S:N501Y. Indeed S:N501Y exhibited a rapid rise and fall in prevalence in Spring 2021 (Fig QQ in S1 File), at the same time as vaccination rates were ramping up in many of the regions in our dataset. Moreover, while the behavior of S:N501Y is partially explained by the rise and fall of B.1.1.7 (Alpha), S:N501Y came to prominence in some regions (notably Brazil) via P.1 (Gamma) where B.1.1.7 was never dominant. Finally, the change in frequency of S:N501Y is correlated to vaccination status (Fig RR in S1 File); this correlation is stronger than the correlation between changes in B.1.1.7 frequency and vaccination status.

S:N501Y thus appears to exhibit vaccination-dependent selection, which explains its relative disappearance from the variant landscape over time as vaccination rates have increased. The precise mechanism underlying this behavior is unclear, but several authors have recently shown that S:N501Y exerts epistatic effects on other mutations by altering their antibody escape properties [49–51]. We hypothesize that S:N501Y serves as a linchpin residue within the RBD that constrains the possibilities for vaccine escape when present.

## 4.7 Epistasis

Mounting experimental evidence for epistasis in SARS-CoV-2 [49, 52] raises the question whether these kinds of effects can be inferred from genomic surveillance data. Doing so is expected to be difficult a priori due to the combinatorially large space of possible interaction effects coupled with the fact that many combinations of mutations are unobserved in available

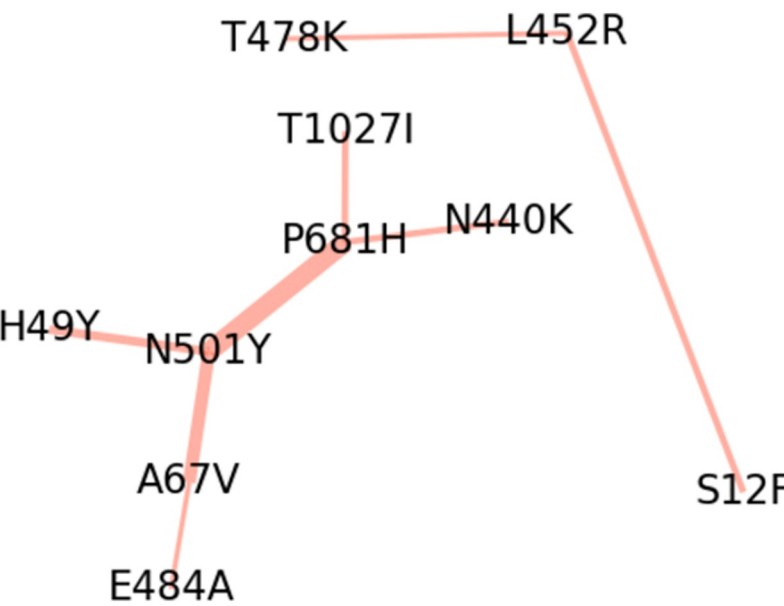

**Fig 11. Top pairwise selection effects.** We depict the two interaction networks of pairwise selection effects in Spike inferred by BVAS. Edge widths are proportional to the posterior inclusion probability of the corresponding pairwise effect. Spatial orientation has no meaning.

data. In this context we expect that explicit sparsity assumptions and high-fidelity statistical inference—key selling points of BVAS—are likely to be crucial.

Here we report initial results of such an analysis, focusing on pairwise interaction effects between non-synonymous mutations in the Spike protein. In particular we consider pairwise interactions between the 421 S mutations in our main analysis—after excluding D614G because it became fixed early in the pandemic—which corresponds to 88410 pairwise interactions. We further exclude pairs of mutations that are not observed together in at least two SARS-CoV-2 clusters—yielding 1432 pairwise interactions—and use BVAS to jointly infer selection effects for the resulting 2975 linear and 1432 pairwise effects. See Sec. S14.9 in S1 File for details.

We summarize our results as a pair of 'interaction networks' in Fig 11 and report more detailed results in Table F in S1 File. We find that top-scoring interaction effects in our analysis are enriched for interactions between mutations that correspond to top-scoring (linear) effects in our default analysis ($P \leq 10^{-4}$). Put differently, we find limited evidence for interaction effects between pairs of mutations where one or both mutations are inferred as approximately neutral on their own.

The interaction effect with the largest PIP is between N501Y in the RBD and P681H adjacent to the furin cleavage site. These two mutations appear together in BA.1/B.1.1.7, while N501Y appears without P681H in P.1/B.1.351 and P681H appears without N501Y in B.1.1.159/B.1.243 (Table F in S1 File). As such there is a plausible evidential basis for inferring the interaction between N501Y and P681H, with each combination of these amino acids appearing in at least 60k SARS-CoV-2 sequences in our dataset. The importance of N501Y in epistatic interactions is concordant with data from [49], who found that N501Y caused the largest shifts in the effects of mutations at other sites using deep mutational scanning libraries of RBD and measuring the impact of every amino acid mutation on ACE2-binding affinity.

Among subleading hits, we find a less well resolved picture due to linkage between different pairs of amino acid mutations, which is reflected in the more moderate PIP scores. For example, all SARS-CoV-2 clusters in our data that carry the pair of amino acids (N501Y, A67V) also carry the pairs (N501Y, P681H) and (A67V, E484A)—but not vice versa. This motivates the graphical representation in Fig 11, which reflects the multiplicity of several amino acid mutations in top-scoring interaction effects—in particular N501Y, P681H, A67V, and L452R.

We expect that resolving these putative interactions in greater detail necessitates follow-up in the lab. Mechanisms of epistasis are likely pleiotropic, but some interactions likely arise from a need to a) incorporate mutations that confer immune escape against the current landscape of circulating variants while b) maintaining protein structure and function. This behavior has been demonstrated for the recurrent S:69–70 deletion [53]. Because the space of possible combinations of alleles is very large, most studies have characterized mutations either individually or in the combinations present in common lineages. By ranking combinations of mutations according to evidence of their selective effects, our model may help focus experimental design in this challenging combinatorial setting.

## 5 Discussion

Bayesian Viral Allele Selection unifies and improves upon the two available methods for inferring selection from large scale genomic surveillance data. One of its strengths is that it makes clear assumptions: i) most alleles are neutral; and ii) viral dynamics is governed by an intuitive discrete time branching process. Other advantages of BVAS include its robustness to hyperparameter choices, its satisfactory uncertainty estimates and the fact that it offers Posterior Inclusion Probabilities. Moreover, the diffusion-based likelihood in Eq (9) is robust to a number of sources of possible bias, including varying sampling rates across time and space and changes in fitness that affect all lineages equally (due to e.g. lockdown measures or variable temperature/humidity).

We highlight the following limitations and refer the reader to Sec. S12 in S1 File for additional discussion. Estimating the effective population size $\nu$ is challenging, especially since $\nu$ can exhibit significant variability across time. While we have argued that sensitivity to $\nu$ is fairly moderate, improved $\nu$ estimates should lead to improved statistical efficiency, especially if $\nu$ can be estimated with finer spatial and temporal granularity. Doing so would likely require incorporating additional sources of data (e.g. case counts) and represents an important direction for future work. Several of the simplifying assumptions that underly BVAS are expected to be violated at some level in real world data. Notably, our basic fitness model is unable to account for epistasis (see Eq (2)). While we have extended this linear model to include pairwise interactions, we have limited this analysis to the Spike protein. Because a genome wide epistasis analysis must contend with millions of possible interactions a priori, additional assumptions to reduce the space of selection effects considered are likely required to make this kind of analysis statistically and computationally tractable. For example, one might limit the analysis to pairs of mutations that are near each other in space.

In summary, BVAS provides a principled statistical and computational framework to identify selection under the constraint of sparsity. Applying BVAS to 6.9 million SARS-CoV-2 genomes provides a detailed picture of viral selection in action. We anticipate that BVAS will be widely applicable to SARS-CoV-2 and other viruses as large scale genomic surveillance data become increasingly available.

## Supporting information

**S1 File. Supplementary figures, tables, and methodological details.** This supplementary pdf contains Figures A-TT and Tables A-F as well as detailed descriptions of our data processing pipeline, modeling assumptions, and inference approach.
(PDF)

**S2 File. Allele-level inference results.** This csv file contains allele-level inference results for our main BVAS analysis utilizing data collected through April 18th, 2022.
(CSV)

**S3 File. Lineage-level inference results.** This csv file contains lineage-level inference results for our main BVAS analysis utilizing data collected through April 18th, 2022.
(CSV)

## Acknowledgments

We gratefully acknowledge colleagues from the originating laboratories responsible for obtaining SARS-CoV-2 specimens. Likewise we gratefully acknowledge colleagues from the submitting laboratories where genetic sequence data were generated and shared via the GISAID initiative. This research would not be possible without their collective efforts; see data availability statement for more information on the data used. We warmly thank Angie Hinrichs for providing the UShER tree that forms a key component of our data pre-processing pipeline. This work would not be possible without her gracious assistance. We also thank Nikolaos Barkas, Stephen F. Schaffner, Jesse D. Pyle, Lonya Yurkovetskiy, Matteo Bosso, Daniel J. Park, Mehrtash Babadi, Bronwyn L. MacInnis, Jeremy Luban, and Pardis C. Sabeti for discussions about SARS-CoV-2.

## Author Contributions

**Conceptualization:** Martin Jankowiak.

**Data curation:** Martin Jankowiak, Fritz H. Obermeyer.

**Formal analysis:** Martin Jankowiak.

**Investigation:** Martin Jankowiak, Jacob E. Lemieux.

**Methodology:** Martin Jankowiak.

**Project administration:** Martin Jankowiak.

**Resources:** Martin Jankowiak.

**Software:** Martin Jankowiak, Fritz H. Obermeyer.

**Supervision:** Martin Jankowiak.

**Validation:** Martin Jankowiak.

**Visualization:** Martin Jankowiak.

**Writing – original draft:** Martin Jankowiak.

**Writing – review & editing:** Martin Jankowiak, Fritz H. Obermeyer, Jacob E. Lemieux.

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
