## [Decision Letter · Decision Letter 0]

15 Sep 2022

Dear Dr Jankowiak,

Thank you very much for submitting your Research Article entitled 'Inferring selection effects in SARS-CoV-2 with Bayesian Viral Allele Selection' to PLOS Genetics.

The manuscript was fully evaluated at the editorial level and by independent peer reviewers. The reviewers appreciated the attention to an important topic but identified some concerns that we ask you address in a revised manuscript.

We therefore ask you to modify the manuscript according to the review recommendations. Your revisions should address the specific points made by each reviewer.

[LINK]

Yours sincerely,

Takashi Gojobori

Academic Editor

PLOS Genetics

Bret Payseur

Section Editor

PLOS Genetics

Reviewer's Responses to Questions

**Comments to the Authors:**

Reviewer #1: A most interesting, well written and presented paper. I have only a few minor comments and suggestions.

Line 28: Add computationally before expensive (assuming this is what you mean) and remove from line 30.

Line 53: Give a brief I sentence definition of neutral/non-neutral alleles as this is an important aspect of the paper.

Lines not given: In methods the sentence begging ‘Since SARS-CoV2 and SARS-CoV-2 are know…’ Split into two sentences as is somewhat hard to read in its current format.

Lines 390-391: Spurious alleles you mean not SARS-CoV-2, just be more specific.

Line 444/Table1: BA4 does not actually appear in the table so remove from the sentence or add to the table.

Table3/Lines469: would be useful to have a column specifying, for those mutations in spike, where in the spike protein they are located. Either as simple as RBD/non-RBD or more specific as you pick up on the FC site later.

Line 500: does ‘S-gene’ include RBD or is it exclusive of RBD given that RBD has it’s own category.

4.4 Backtesting: Would be interesting to include omicron BA.4 and BA.5 analysis here, if it is possible to do in a timely manner. This would also make the paper more relevant to the current COVID-19 landscape.

Reviewer #2: In this manuscript, Jankowiak et al. present a scalable probabilistic method to identify the relative fitness and growth rate of viral lineages that can be applied to a global scale of SARS-CoV-2 sequence data. For a statistical method, BVAS can handle an incredible scale while maintaining high accuracy. The authors did a commendable job in rigorously developing and evaluating this method. The scalability of BVAS is owed to simple and realistic assumptions that most alleles are neutral and can be modeled using a discrete time branching process. On simulated data, BVAS outperforms state-of-the-art in both scale and accuracy. On real SARS-CoV-2 data, it is able to rank variants that match well with experimental data. With backtesting, the authors show that BVAS can be useful to identify concerning SARS-CoV-2 lineages early. The authors also present intriguing results in evaluating the epistatic interactions of mutations using their approach.

In my opinion, the manuscript is well-written and publishable in its current form. I have the following minor comments and questions for the authors:

1. Sequencing rates and surveillance efforts are highly non-uniform across different nations. Perhaps the author should discuss how this may affect BVAS and how realistic is its assumption of an i.i.d. sampling rate.

2. Simulated datasets use a sampling rate between 1-64%. It is likely that the COVID-19 sampling rate is a lot lower than 1%. Why not evaluate the results for smaller sampling rate values as well?

3. Simulated dataset use a small reproduction number range of 0.9 to 1.1. The range of R0 for SARS-CoV-2 is much higher. How does BVAS perform when this range is increased?

4. What is the value of sigma-Laplace for data in Fig. 1.?

5. For Fig. 4, why is the hit rate not optimal at #non-neutral sites=10?

**Have all data underlying the figures and results presented in the manuscript been provided?**

Reviewer #1: Yes

Reviewer #2: Yes

PLOS authors have the option to publish the peer review history of their article (what does this mean?). If published, this will include your full peer review and any attached files.

Reviewer #1: **Yes: **Alexander Stewart

Reviewer #2: No

---

## [Decision Letter · Decision Letter 1]

23 Nov 2022

Dear Dr Jankowiak,

We are pleased to inform you that your manuscript entitled "Inferring selection effects in SARS-CoV-2 with Bayesian Viral Allele Selection" has been editorially accepted for publication in PLOS Genetics. Congratulations!

Yours sincerely,

Takashi Gojobori

Academic Editor

PLOS Genetics

Bret Payseur

Section Editor

PLOS Genetics

Comments from the reviewers (if applicable):

Reviewer's Responses to Questions

**Comments to the Authors:**

Reviewer #2: The authors have addressed my questions and concerns satisfactorily. It might be worth including the author response to my question on the reproduction number range used in simulations in the manuscript.

**Have all data underlying the figures and results presented in the manuscript been provided?**

Reviewer #2: Yes

PLOS authors have the option to publish the peer review history of their article (what does this mean?). If published, this will include your full peer review and any attached files.

Reviewer #2: No

**Data Deposition**

http://datadryad.org/submit?journalID=pgenetics&manu=PGENETICS-D-22-00711R1

**Press Queries**

---

## [Editor Report · Acceptance letter]

7 Dec 2022

PGENETICS-D-22-00711R1 

Inferring selection effects in SARS-CoV-2 with Bayesian Viral Allele Selection 

Dear Dr Jankowiak, 

We are pleased to inform you that your manuscript entitled "Inferring selection effects in SARS-CoV-2 with Bayesian Viral Allele Selection" has been formally accepted for publication in PLOS Genetics! Your manuscript is now with our production department and you will be notified of the publication date in due course.

With kind regards,

Zsofi Zombor

PLOS Genetics

On behalf of:
